# Survivin Expression in Placentas with Intrauterine Growth Restriction

**DOI:** 10.3390/biomedicines13071576

**Published:** 2025-06-27

**Authors:** Pavo Perković, Sanja Štifter-Vretenar, Marina Perković, Marko Štefančić, Ena Holjević, Andrea Dekanić, Tea Štimac

**Affiliations:** 1Department of Gynecology and Obstetrics, University Hospital Merkur, Zajčeva 19, 10000 Zagreb, Croatia; 2Department of Pathology, Aarhus University Hospital, Palle Juul-Jensens Blvd. 161, 8200 Aarhus, Denmark; stifter.sanja@gmail.com; 3Special Hospital for Pulmonary Diseases, Rockefellerova 3, 10000 Zagreb, Croatia; mpti2007@gmail.com; 4Department of Gynecology and Obstetrics, Sestre Milosrdnice University Hospital Center, Vinogradska 29, 10000 Zagreb, Croatia; markostefancic1234@yahoo.com; 5Department of Pathology and Cytology, Clinical Hospital Centre Rijeka, Krešimirova ulica 42, 51000 Rijeka, Croatia; holjevic.ena@gmail.com (E.H.); andrea.dekanic@medri.uniri.hr (A.D.); 6Department of Gynecology and Obstetrics, Clinical Hospital Centre Rijeka, Krešimirova ulica 42, 51000 Rijeka, Croatia; tea.stimac2@gmail.com

**Keywords:** apoptosis, intrauterine growth restriction, placenta, survivin

## Abstract

**Background/Objectives:** Intrauterine growth restriction (IUGR) is a pathological condition defined by a reduced fetal ability to achieve the genetically expected growth potential during gestation. It affects 5–10% of all pregnancies and it is a leading cause of perinatal morbidity and mortality. During the initial phases of placentation, complex interlinked processes including cell proliferation, differentiation, apoptosis and the invasion of trophoblasts occur. Alterations in the regulation of these processes lead to placental dysfunction. Survivin, a member of the inhibitor of apoptosis (IAP) family, plays an important role in cell proliferation balance and apoptosis, thus leading to proper placental development. This study aimed to evaluate survivin expression in placentas from IUGR and healthy pregnancies to explore its potential as a biomarker for the early diagnosis, prevention, and treatment of IUGR. **Methods**: Survivin presence was determined in 153 archival formalin-fixed and paraffin-embedded placental tissues from IUGR (N = 122) and uncomplicated (N = 31) term pregnancies. Tissue microarrays (TMAs) were constructed, and survivin expression was assessed using immunohistochemistry (IHC). Survivin levels were quantified using positive cell proportion (PCP) scores and immunoreactive scores (IRS), with statistical significance determined using mean values, standard deviation (SD), standard error, and Student’s t test in instances of normal distribution, and when this was not the case, the Mann–Whitney test. Chi-square tests, Fisher exact tests, and *t*-tests (*p* < 0.05) were used to compare categorical variables. **Results**: Our results suggested the significantly higher expression of survivin validated with PCP (*p* < 0.001) and IRS (*p* < 0.002) in placentas with IUGR compared to placentas from non-complicated term pregnancies. **Conclusions**: Increased survivin expression in IUGR placentas points to its potential role as a key indicator of placental dysfunction. By signaling early pathological changes, survivin may offer a valuable tool for the early detection of IUGR, potentially allowing for timely clinical interventions that could reduce the risk of serious outcomes, including stillbirth. To fully establish survivin’s clinical value, further research is needed to validate its diagnostic accuracy and to explore its involvement in molecular pathways that may be targeted for therapeutic benefit.

## 1. Introduction

Intrauterine growth restriction (IUGR) is a significant contributor to perinatal morbidity and mortality worldwide, affecting approximately 5–10% of pregnancies [1]. IUGR is defined as impaired fetal growth velocity secondary to placental insufficiency, resulting in the failure of the fetus to reach its genetic growth potential [2]. The placenta is central to fetal development, and relies on tightly regulated trophoblast proliferation, differentiation, fusion, and invasion to properly establish the maternal–fetal interface. In IUGR, placental dysfunction is marked by defective trophoblast fusion and syncytialization, leading to compromised syncytiotrophoblast formation and impaired nutrient and gas exchange [3]. Normal placental development requires a balance between proliferation, apoptosis, angiogenesis, and vascular remodeling. The disruption of these processes can result in an abnormal placental structure, uteroplacental vascular insufficiency, increased resistance, and a reduced placental size, all of which are observed in IUGR [4,5,6,7,8]. These pathological changes reduce the delivery of fetoplacental nutrients and oxygen and produce the hypoxia and undernutrition that define IUGR [7]. Doppler ultrasound studies confirm that abnormal umbilical and uterine artery blood flow—such as absent or reversed end-diastolic flow—is associated with adverse pregnancy outcomes in IUGR [9,10]. Survivin (BIRC5), a dual-function member of the IAP family, has recently received attention for its role in placental physiology and pathology. Survivin is a potent inhibitor of apoptosis, and also promotes cell proliferation, mediates mitosis, and supports trophoblast invasion via the modulation of cell cycle regulators and matrix metalloproteinases [11,12]. In healthy pregnancies, survivin is highly expressed in cytotrophoblasts and is crucial for preserving placental tissue integrity and supporting normal differentiation. Recent studies demonstrate that survivin expression is reduced in preeclampsia and IUGR, correlating with increased trophoblast apoptosis and invasive defects, while overexpression is seen in gestational trophoblastic disease [13]. However, survivin’s precise spatiotemporal expression dynamics in IUGR placentas have not been fully delineated, limiting its current utility as a biomarker or therapeutic target.

The goal of this study was to evaluate the expression of survivin in placental tissues from both uncomplicated pregnancies and those complicated by IUGR. A more thorough understanding of survivin’s role in placental dysfunction may help identify novel biomarkers and therapeutic strategies for the early recognition and management of IUGR.

## 2. Materials and Methods

### 2.1. Patients and Placental Tissue Samples

This study was conducted on 153 archival formalin-fixed and paraffin-embedded placental tissues from the Department of Pathology and Cytology at Clinical Hospital Center Rijeka, collected during delivery at the Department of Obstetrics and Gynecology of the same hospital. Samples were divided into groups of placentas (N = 122) from term pregnancies (≥37 weeks of gestation) complicated with IUGR and were compared with placentas (N = 31) from uncomplicated term pregnancies, controlled for basic epidemiological factors. IUGR was diagnosed when the fetal birth weight was <10th percentile and there was evidence of uteroplacental insufficiency detected by Doppler velocimetry. Not every fetus <10th percentile is pathologically small; evidence of placental dysfunction (abnormal Dopplers, amniotic fluid, growth velocity) is required to confirm FGR, rather than the fetus being constitutionally small (SGA). Doppler velocimetry consisted of umbilical artery and middle cerebral artery measurements, and Resistance index (RI) and Pulsatility index (PI) measurements. Also, the Cerebroplacental ratio (CPR) was calculated, and CPR <5th percentile or umbilical artery PI >95th percentile were used as diagnostic criteria. Some ancillary findings support IUGR diagnosis, like oligohydramnios detected by amniotic fluid index pregnancies (≥37 weeks of gestation) (AFI) <5 cm or a single deepest pocket <2 cm [14,15]. The weight of all placentas was noted. The exclusion criteria included multifetal pregnancy, intrauterine viral infections, chorioamnionitis, maternal chronic arterial hypertension or autoimmune diseases, fetal anomalies or chromosomal abnormalities, and stillbirth.

As positive controls, archival samples of placental samples from the first trimester of normal pregnancies were used. Negative controls included tissue sections processed identically but with the omission of the primary antibody.

The Ethics Committee of the institutional review board at Rijeka University Hospital Center approved the investigation.

### 2.2. Tissue Selection and Regional Standardization

To account for potential regional heterogeneity in survivin expression, placental sampling followed a standardized protocol:

Macroscopic Examination: Each placenta was sectioned to identify the central (near cord insertion) and peripheral (marginal) zones, basal plate (maternal interface), and chorionic plate (fetal surface).Coring Strategy: Two 2 mm cores were extracted per placenta:Core 1: Central region (within 2 cm of umbilical cord insertion).Core 2: Peripheral region (1 cm from the placental margin).

Hematoxylin–eosin-stained sections from each core were reviewed by two pathologists to verify the presence of villous trophoblasts and exclude infarcts, calcifications, or non-representative areas. For the basal and chorionic plates, one additional core per region was included in a subset of samples (n = 30) to assess regional variability. Placental sampling targeted the central and peripheral regions to ensure representative trophoblast analysis. Histologically confirmed cores excluded infarcts or calcifications.

### 2.3. TMA Construction and Immunohistochemical Semi Quantification

From representative samples of whole placental tissue stained with hematoxylin–eosin (HE), representative areas were selected and labelled with appropriate paraffin blocks for tissue microarray (TMA) construction. Two tissue cores, each 2 mm in diameter, were placed in a recipient paraffin block using an MTA Booster OI manual tissue arrayer (Alphelys, Plaisir, France). The final TMA blocks contained 306 cores with tissue specimens. The cores were spaced at intervals of 0.5 mm in the x- and y-axes. One section from each TMA block was stained with HE for morphological assessment. Serial sections were cut from TMA blocks for immunohistochemical staining. In each IHC run, positive controls consisted of non-pathological placental tissue with established survivin expression, while negative controls were performed by omitting the primary antibody on parallel tissue sections. Normal kidney tissue was included in TMA blocks solely for orientation purposes and not as a control for survivin staining. Five-micrometer-thick sections were placed on adhesive glass slides (Capillary Gap Microscope Slides, 75 μm, Code S2024, Dako Cytomation, Glostrup, Denmark), allowed to dry overnight at 37 °C, and stored in darkness at +4 °C.

To minimize variability, all TMAs were stained using the same protocol, reagents, and antibody lots. Positive and negative controls were included in each run, and representative sections from each TMA batch were processed in parallel. No staining or batch effect variability was observed between TMAs.

### 2.4. Analysis of Results

The quantification of the analyzed protein Anti-Survivin antibody ab469 by Abcam was performed by analyzing scans of IHC-stained samples. PCP was scored as follows: 0 (no positive cells), 1 (<10% of positive cells), 2 (10–50% positive cells), 3 (51–80% positive cells), and 4 (>80% positive cells). The staining intensity was categorized as follows: 0 (no color reaction), 1 (mild reaction), 2 (moderate reaction), and 3 (intense reaction). The IRS ranged from 0 to 12, calculated by multiplying the positive cells proportion score and staining intensity score (SIS) (0–3). Scores were classified as follows: 0–1 (negative), 2–3 (mild), 4–8 (moderate), and 9–12 (strongly positive).

Two pathologists independently scored the survivin-stained slides. Any scoring differences were resolved by joint review and agreement.

In our study, the cutoffs for mild, moderate, and strong expression were chosen based on previously published studies for consistency in the visual assessment [16,17].

### 2.5. Statistical Analysis

For the statistical analysis of data, we used mean values, standard deviation (SD), standard error and Student’s *t* test in instances of normal distribution, or, when that was not the case, the Mann–Whitney test. The chi-square test and Fisher’s exact test were used to compare categorical variables. Statistical significance was defined as a *p*-value less than 0.05.

## 3. Results

### Demographic Data of Study Population

The study included 122 placentas from IUGR pregnancies. The control group included 31 placentas from normal, healthy term pregnancies without pathologies. The matching of both groups in terms of their clinical characteristics is presented in Table 1.

Figure 1 shows IUGR placental tissue after sampling and staining for anti-survivin antibodies. Two independent pathologists quantified anti-survivin antibodies by analyzing stained samples. They compared PCP and IRS in placentas with IUGR to those from healthy pregnancies.

A significantly higher expression of survivin, validated by PCP (*p* < 0.001) and IRS (*p* < 0.002), was observed in placentas with IUGR compared to those from healthy controls (Table 2).

## 4. Discussion

IUGR affects one out of ten pregnancies and is strongly associated with stillbirths and severe perinatal morbidity [18,19]. Undetected IUGR is responsible for up to 50% of preventable stillbirths [20,21]. Today, there remains no effective treatment for IUGR, and management is based on the ultrasound detection of small fetuses that do not reach their genetic growth potential. In addition to assessing fetal size, ultrasound is essential for evaluating the amniotic fluid volume and placental morphology. Doppler velocimetry, particularly of the umbilical artery, middle cerebral artery, and ductus venosus, provides critical information about uteroplacental and fetal circulation. Current management strategies are supportive, still not curative, and mainly rely on timely delivery [14,15,22]. Timely delivery is a balance between fetal morbidity, mortality and prematurity [22]. Currently, there is no reliable screening test for IUGR. Many diagnoses are made too late, and a non-invasive biomarker for early detection reflecting placental insufficiency would be ideal. The application of biomarkers could also be an interventional therapeutic for the timely prevention of the disease, as well as the monitoring of therapies [1].

Survivin is a member of the IAP family. Survivin inhibits caspase activation, which is a proteolytic component of the apoptosis (programmed cell death) pathway and leads to the negative regulation of apoptosis [23]. Survivin was initially identified in human placenta by Lehner et al. and is important for early embryogenesis due to its role in regulating physiological trophoblast proliferation [23,24,25]. Placenta development during pregnancy involves a dynamic process of trophoblast turnover, which consists of proliferation, differentiation, fusion, and apoptosis [26,27]. Evidence indicates that apoptosis mainly occurs in syncytiotrophoblasts (STBs) rather than in cytotrophoblasts, and this appears to increase as gestation advances [3]. Survivin is nearly absent in normal adult tissues but highly expressed in various cancers [28]. It is now extensively studied in oncology due to its associations with chemotherapy resistance, increased tumor recurrence and heightened tumor aggressiveness, its role as a prognostic factor for cancer patients, and its potential as a therapeutic target for cancer treatment [28]. Apoptosis plays a significant role in normal placental development and remodeling. During early placental development and gestation, properly regulated apoptotic mechanisms facilitate proper tissue remodeling and vascular adaptation. Disturbances in feto-placental apoptosis could cause abnormal, pathological placental development with compromised placental function. This insufficient placental function, which favors increased apoptosis and reduced proliferation, compromises the development of placental villi and fails to support optimal fetal growth, leading to pregnancy complications such as IUGR [23]. The regulation of apoptosis represents a critical factor in the pathogenesis of placental disorders that contribute to adverse perinatal outcomes.

Our results showed a significantly higher expression of surviving, validated with PCP (*p* < 0.001) and IRS (*p* < 0.002), in placentas with IUGR compared to placentas from healthy controls. In addition to survivin’s established role in inhibiting apoptosis and regulating cell proliferation, the complex molecular mechanisms involved are still not fully understood. To date, only a limited number of studies on the role of survivin in IUGR have been published, with the findings inconsistent across the available literature [22,23,25]. An investigation examining survivin in cord blood at term found no significant differences between pregnancies with IUGR, large for gestational age (LGA) pregnancies and appropriate for gestational age (AGA) pregnancies [25]. The lack of significant differences suggests that cord blood survivin levels may not represent the disturbances in feto-placental apoptosis, and the authors suggested the need for additional studies that will report the placental expression of survivin [25]. In addition to the few studies associating survivin with IUGR, other studies have explored survivin expression in the context of other placental pathologies, such as preeclampsia. A study published over ten years ago found that the placental expression of survivin decreases with the severity of preeclampsia, indicating that the dysregulation of apoptosis-related proteins could be a common finding within various forms of placental pathologies [23]. The authors concluded that a lower expression of survivin is observed in severe preeclamptic placentas. They suggested that the expression of survivin may be correlated with the severity of preeclampsia [23]. They found also that the level of survivin decreased with gestational age [23]. They highlighted the role of apoptosis in preeclampsia and IUGR, focusing on how disrupted cytotrophoblast turnover can release trophoblastic material into maternal circulation, causing endothelial cell activation, a key symptom of preeclampsia [23]. The molecular mechanisms causing preeclampsia and IUGR are not well understood. It is known that they are not the same, as only some pre-eclamptic patients also have IUGR. Also, only a few IUGR patients will have symptoms of endothelial disturbances, which are typical in preeclampsia. Moreover, in the study group, there were only eight preeclamptic women with IUGR among all the patients, the criteria for the diagnosis of IUGR was not clear, and they probably had different gestational ages [23]. Whitehead et al. found a significantly higher level of survivin expression in the placenta and maternal blood, with a progressive increase as the severity of IUGR increased [22]. They commented upon the conflicting results of other studies that confused IUGR with fetuses that were small for their gestational age, because all participants in their study were assessed with Doppler velocimetry, which ensured that placental insufficiency was present, upregulating both pro-and anti-apoptotic genes (and thus can cause an increase in the survivin level) as a response to hypoxia [18]. They concluded that the disordered regulation of apoptosis in placental dysfunction is not yet clearly understood. However, circulating mRNA-coding genes that regulate apoptosis may serve as useful biomarkers for assessing placental function, potentially enabling clinicians to identify early-onset IUGR. In our study, we clearly differentiate IUGR from SGA, and found high survivin expression in IUGR placentas, supporting previous findings [22]. The strengths of our study include the large number of patients in the study group and the use of well-established criteria for the diagnosis of IUGR, but it can, of course, be improved by the more objective and quantitative measurement of survivin in placental tissue and other samples, like umbilical or maternal blood.

Survivin, as a part of the IAP family, prevents programmed cell death by blocking caspases and could prevent proper trophoblast invasion into the uterine wall, which causes placental insufficiency leading to IUGR [3,19]. Survivin is involved in mitotic regulation in the cell cycle, where overexpression can cause mitotic errors due to improper chromosome segregation during mitosis. Also, survivin could be involved in signaling pathways that promote excessive cell proliferation or implicated in angiogenesis because it enhances endothelial cell survival in response to an imbalance of growth factors, such as vascular endothelial growth factor (VEGF), that interact with survivin [3,25,27]. Many of these functions are well known and have been investigated extensively in oncology, and we can establish some correlation between tumor invasion and trophoblast invasion, which are necessary for proper placenta development [19,28,29,30]. Also, it seems that hypoxia caused by poor vascular development, which is often present and diagnosed in placental insufficiency and IUGR, can influence survivin levels. This could explain the conflicting results in a small number of studies about survivin expression in IUGR placentas. The available literature suggests a lack of understanding regarding survivin’s role in placental disorders, including IUGR. More comprehensive studies aimed at understanding the underlying apoptotic signaling pathways and performing molecular analyses of placental tissues are needed. This type of research would contribute to a clearer understanding of the molecular mechanisms underpinning IUGR.

In conclusion, survivin could be used as a biomarker (non-invasive diagnostic test) for the screening and moderation of antepartum care for the timely delivery and prevention of stillbirth. It could also be a target for preventative or therapeutic interventions aimed at survivin-related pathways. Future validations with large prospective studies are required to establish the role of survivin in IUGR.

## Figures and Tables

**Figure 1 biomedicines-13-01576-f001:**
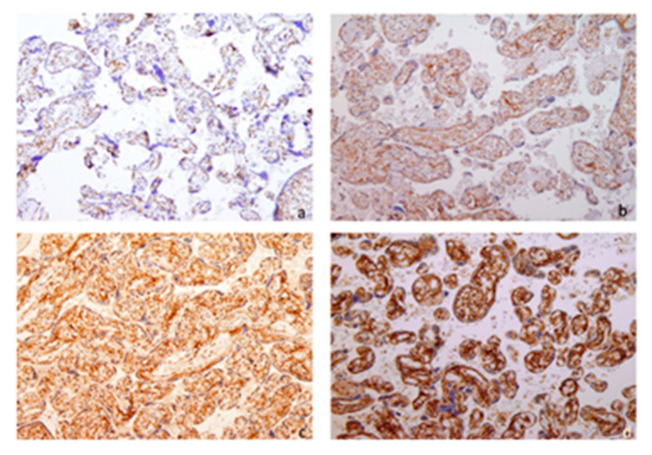
IUGR placental tissue, immunohistochemical staining for anti-survivin antibodies (cytoplasmic staining), 10× magnification: (**a**) negative staining, (**b**) staining intensity 1, (**c**) staining intensity 2, (**d**) staining intensity 3.

**Table 1 biomedicines-13-01576-t001:** The demographic and clinical data of women with IUGR and the control group.

	IUGR N = 122	Control Group N = 31	*p*
Maternal age (years)	35.3 (5.7) *	33.0 (4.0)	0.042
Parity (% primiparous)	58.2	54.8	
Gestational age at delivery (weeks)	39.3 (1.1)	39.7 (1.05)	0.123
Birth Weight (grams)	2542 (255)	3487 (402)	*p* < 0.001
Male fetal gender (%)	43.4	45.2	

* Mean (SD).

**Table 2 biomedicines-13-01576-t002:** Staining intensity score and immunoreactive score for survivin in placental tissue.

Survivin	IUGR N (%)	Control Group N (%)	X^2^ Value	df	*p*
Staining intensity score *			12.594	1	<0.001
1	10 (8.2)	10 (32.3)			
2	74 (60.7)	15 (48.4)			
3	38 (31.1)	6 (19.4)			
Immunoreactive score **			12.783	1	0.002
4	10 (8.2)	10 (32.3)			
8	74 (60.7)	15 (48.4)			
12	38 (31.1)	6 (19.4)			

* The Staining intensity score: negative (0), weak (1), moderate (2), strong (3); ** The Immunoreactive score is a combination of the intensity and proportion of positive cells; eight points or more is regarded as IRS-positive.

## Data Availability

Data are contained within the article.

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
