# Peer review of "Survivin Expression in Placentas with Intrauterine Growth Restriction"

_biomedicines, 2025, doi:10.3390/biomedicines13071576_

Round 1
Reviewer 1 Report
Comments and Suggestions for Authors
Dear Authors
Thank you for inviting me to review your manuscript on survivin expression in IUGR placentas. The topic is highly relevant as it deals with a major pathmechanism of IUGR and postulates survivin as a biomarker. The investigation has been designed well but certain major issues need to be addressed so as to enhance its scientific rigor and clinical utility
These are my thorough suggestions for major revision:
1. Hypothesis Explanation and Objectives
1.1 The introductory sentence mentions the potential of survivin as a biomarker, but fails to elucidate why survivin was chosen, in particular, among so many other regulators of apoptosis.
1.2 Authors wrote: “Placental dysfunction associated with placental development accountable for fetal growth in IUGR presents with defective trophoblast fusion and syncytialization.” To prove this statement i recommend referencing following paper on placental flow in low and high risk pregnancy:
- https://doi.org/10.1080/14767050600852510 and DOI: 10.1055/a-2075-3021
2. Study Group Definition and Selection
2.1 The definition of IUGR only depends on birth weight <10th percentile and on Doppler results, but you do not specify what Doppler parameters were used (e.g., umbilical artery PI, MCA, ductus venosus, CPR).
2.2 Provide diagnostic criteria and list a reference used for IUGR classification.
3. Placental Sampling Strategy
3.1 Provide more detail on sampling standardization. Were samples taken from separate placental regions (central, peripheral, basal plate, chorionic plate)? It matters because survivin expression may vary between regions.
3.2 Was result sampling blinded to reduce selection bias?
4. Omitted Methodological Information
4.1 There are no data on interobserver agreements between both pathologists.
4.2 Be more descriptive about IHC negative and positive controls. Were non-pathological pregnancy placental tissue utilized as a positive control or was kidney tissue used only as an orientation control?
4.3 State if staining variability or batch effect between TMAs was addressed or corrected
5. IRS and PCP Scoring System Justification 5.1 The IRS placental tissue grading into mild, moderate, and strong expression has no firm validation and arbitrarily refers to thresholds. Justify or cite a previous source with thresholds employed. 5.2 Think about if IRS is sensitive and specific enough for your purpose, or if another quantification (i.e., H-score or digital image analysis) should be noted or recommended
6. Results Presentation
6.1 Stratify IUGR into mild and severe based on percentiles (<3rd vs. 3rd-10th) and examine survivin expression between these subgroups.
Best regards
Reviewer 2 Report
Comments and Suggestions for Authors
This manuscript aims to demonstrate that survivin is a good biomarker for IUGR. Survivin is an important anti-apoptotic, pro-proliferation protein. The authors use a large sample of IUGR tissue and appropriate controls. The establishment of tissue arrays with these samples is important to control for technical differences. This removes the variation that can occur if different tissues on different slides are assayed in separate experiments.
Despite the positives mentioned above, the paper lacks sufficient data. One figure and two tables with little detail are not sufficient. Also the introduction is incomplete.
You saw a correlation with IUGR and survivin, but to strengthen this observation it would be good to have controls, both positive and negative. You should look at other proteins that should correlate with IUGR and also control proteins that should not correlate.
You could also use the tissue arrays to do high thoughput analysis. Imaging Mass Cytometry would allow you to look at multiple proteins at once. Spatial transcriptomics (RNA-seq) can be done on the tissue microarrays (ref: NOT my work....Juwayria, Shrivastava, P., Yadav, K. et al. Microarray integrated spatial transcriptomics (MIST) for affordable and robust digital pathology. npj Syst Biol Appl 10, 142 (2024). https://doi.org/10.1038/s41540-024-00462-1). This would give you a huge data base that would help validate the survivn data and reveal other biomarkers.
Reviewer 3 Report
Comments and Suggestions for Authors
Thank you for giving me the access to review this article entitled: Survivin expression in placentas with intrauterine growth restriction .firstly the paper is important and novel but need many answers to some questions
Abstract
Line 20 correct the error in the sentence.
Line 24 please change the sentence plays a crucial role to play an important role in cell proliferation balance.
Line 35 non-complicated term pregnancies. Is normal ones?
What about changes occur in non complicated pregnancy?
What about statical analysis done in the abstract part.?
Please rephrasing the conclusion part of the abstract to be more clear.
Keywords must be reorganized with first capital letter and alphabetical order.
Introduction
Line 47 this paragraph:Placental dysfunction, associated with the development of the placenta which is essential for fetal growth, is characterized by impaired trophoblast fusion and syncytialization in IUGR.this paragraph need more relevant references.
Last paragraph in introduction section should be reorganized and rephrased again to be more clear
Materials
Line 67 what about ethical approved number?
What about study limitation?
How authors calculate sample size?
Line 73 how authors determine utero placental insufficiency detected by Doppler?
Line 81-68 this paragraph need to be summarized with references
Line 91 what about statiscal analysis?.please mention the statiscal test that was performed with determination of sample size
Discussion
Line 134 more information was needed regarding the effective treatment of IUGR,with the role of ultrasound analysis.
Line 155-170 need to be rephrased again please justify.
Line 199 please check reference number 19
Figure 1 need more bigger size of the image with more resolution
Comments on the Quality of English Language
The English language should be reorganized with revision of extra spacing and typing errors
Round 2
Reviewer 2 Report
Comments and Suggestions for Authors
The authors made improvements to the manuscript by the addition of more details and more clarity in the introduction, methods and discussion.
I would like to see more histology results. Figure 1 states placenta staining but not if it is IUGR or control. The reader should not have to assume it is IUGR placenta. Control placenta staining should also be included. Next, there is always bias when analyzing by eye. There are computer programs such as ImageJ that can be used to quantitate different samples without bias.
I still recommend that the analysis of other proteins need to be included in the results. A known biomarker for IUGR. The results should positively correlate with Survivn. Also a negative protein should be analyzed, one that has not been linked to IUGR. This would strengthen your positive results and confirm that the correlation is not technical. FIxation of tissue can cause random staining. If the controls and the IUGR were fixed by different people at different times this could lead to false positives. Positive and negative controls will resolve the impact of technical based differences.
